# A Retrospective Analysis of Systemic *Bartonella henselae* Infection in Children

**DOI:** 10.3390/microorganisms12040666

**Published:** 2024-03-27

**Authors:** Ramona Florina Stroescu, Flavia Chisavu, Ruxandra Maria Steflea, Gabriela Doros, Teofana-Otilia Bizerea-Moga, Dan Dumitru Vulcanescu, Teodora Daniela Marti, Casiana Boru, Cecilia Roberta Avram, Mihai Gafencu

**Affiliations:** 1Department XI of Pediatrics—1st Pediatric Discipline, Center for Research on Growth and Developmental Disorders in Children, “Victor Babes” University of Medicine and Pharmacy Timisoara, Eftimie Murgu Sq. No. 2, 300041 Timisoara, Romania; stroescu.ramona@umft.ro (R.F.S.); steflea.ruxandra@umft.ro (R.M.S.); 24th Pediatric Clinic, “Louis Turcanu” Children’s Clinical and Emergency Hospital, Iosif Nemoianu 2, 300011 Timisoara, Romania; farkas.flavia@umft.ro (F.C.); doros.gabriela@umft.ro (G.D.); mgafencu@umft.ro (M.G.); 3Centre for Molecular Research in Nephrology and Vascular Disease, Faculty of Medicine “Victor Babes”, “Victor Babes” University of Medicine and Pharmacy Timisoara, 300041 Timisoara, Romania; 4Department XI of Pediatrics—3rd Pediatric Discipline, “Victor Babes” University of Medicine and Pharmacy Timisoara, Eftimie Murgu Sq. No. 2, 300041 Timisoara, Romania; 5Department of Microbiology, “Victor Babes” University of Medicine and Pharmacy Timisoara, Eftimie Murgu Sq. No. 2, 300041 Timisoara, Romania; dan.vulcanescu@umft.ro; 6Multidisciplinary Research Center on Antimicrobial Resistance (MULTI-REZ), Microbiology Department, “Victor Babes” University of Medicine and Pharmacy Timisoara, Eftimie Murgu Sq. No. 2, 300041 Timisoara, Romania; 7Department of Medicine, “Vasile Goldis” University of Medicine and Pharmacy, 310414 Arad, Romania; marti.teodora@uvvg.ro (T.D.M.); boru.casiana@uvvg.ro (C.B.); 8Department of Microbiology, Emergency County Hospital, 310037 Arad, Romania; 9Department of Residential Training and Post-University Courses, “Vasile Goldis” Western University, 310414 Arad, Romania; avram.cecilia@uvvg.ro

**Keywords:** infectious diseases, *Bartonella henselae*, children, pediatrics

## Abstract

Systemic *Bartonella henselae* infection, also known as cat-scratch disease (CSD), presents a diagnostic challenge due to the variability of clinical manifestations and the potential for serological cross-reactivity with other organisms. This study aimed to retrospectively analyze the epidemiological, clinical, laboratory, and imaging characteristics of pediatric patients diagnosed with systemic *B. henselae* infection, to improve understanding and facilitate timely diagnosis and treatment. We conducted a 10-year retrospective study at the “Louis Turcanu” Children’s Emergency Hospital and private clinics in Timisoara, Romania, reviewing records for confirmed cases of *B. henselae* infection from January 2014 to January 2024. The study adhered to the Declaration of Helsinki and received approval from the Institutional Review Board. Diagnostic criteria included contact with animals, prolonged fever, hematological and/or hepatosplenic manifestations, and positive serological tests for *B. henselae*. Nineteen pediatric patients were identified with a median age of 8.1 years. The majority were exposed to felines (94.7%), reflecting the disease’s epidemiological profile. Clinical findings highlighted fever (47.4%), lymphadenopathy (78.9%), and less frequently, abdominal pain and headache (both 10.5%). Laboratory analyses revealed a mean hemoglobin of 12.6 mg/dL, WBC count of 13.1 × 10^3^ cells/microliter, and platelet count of 340.6 × 10^3^ per microliter. Significant findings included elevation in ESR and CRP in 47.4% and 21.1% of patients, respectively, and high seropositivity rates for *B. henselae* IgM (63.2%) and IgG (94.7%). Imaging studies demonstrated widespread lymphadenopathy and occasional splenomegaly and hepatic microabscesses. All patients received antibiotic therapy, with azithromycin being the most commonly used (94.7%). Co-infections with Epstein–Barr Virus, Cytomegalovirus, and *Toxoplasma gondii* were documented, indicating the complex infectious status of the patients. Systemic *B. henselae* infection in children predominantly manifests with fever and lymphadenopathy, with a significant history of exposure to felines. Laboratory and imaging findings support the diagnosis, which is further complicated by potential co-infections. Effective antibiotic therapy, primarily with azithromycin, underscores the need for comprehensive diagnostic and treatment strategies. This study emphasizes the importance of considering systemic *B. henselae* infection in pediatric patients with prolonged fever and contact with cats, to ensure timely and appropriate treatment.

## 1. Introduction

Although rare in immunocompetent children, systemic clinical manifestations of *Bartonella henselae* infection have been documented in a limited number of immunocompromised and immunocompetent children [1]. Systemic bartonellosis should be suspected if there is a prolonged fever and many granulomatous lesions in the liver and spleen that are shown on ultrasonography, computer tomography (CT), or magnetic resonance imaging (MRI) [2]. Clinical manifestations of bartonellosis can be quite varied, ranging from an infection without any symptoms to a widespread illness involving several organs [3,4].

In 85–90% of cases, regional lymphadenopathy is the characteristic clinical sign of *Bartonella* infection, which is a benign self-limiting illness [3,5,6]. About one-third of the time, lymphadenopathy is accompanied by other symptoms such as fever, malaise, headache, nausea, and stomach discomfort. Less frequently, an unusual variant may occur with atypical clinical signs, such as hepatosplenic illness, osteomyelitis, ophthalmic abnormalities, fever of unexplained origin, or neurological involvement with encephalitis or endocarditis [2,7]. An unmonitored atypical *Bartonella* infection may exhibit rapid progression in a short timeframe; as such, rapid diagnosis is important [8].

Due to the lack of gold standard criteria and the variability of clinical presentation, systemic bartonellosis diagnosis is frequently challenging and delayed. Furthermore, it has been reported that serological tests for *B. henselae* exhibit cross-reactivity with other organisms, such as *Bartonella quintana*, *Coxiella burnetii*, and *Chlamydia* spp., which complicates diagnosis [9,10,11].

In light of the complexity in regard to clinical presentation in systemic bartonellosis in children, along with challenges in timely diagnosis and management, the present retrospective analysis aims to contribute with insights into the prevalence, symptomatology, investigations and treatment of this pathology. Also, a secondary aim of this study is to underscore the need for enhanced awareness of Bartonella infections, especially in pediatric patients.

## 2. Materials and Methods

### 2.1. Design and Ethics

A 10-year retrospective analysis was performed between January 2014 and January 2024 at the “Louis Turcanu” Children’s Emergency Hospital from Timisoara, Romania and private clinics. The hospital paper and digital records were searched explicitly for the infection with *B. henselae.* The study was conducted in accordance with the Declaration of Helsinki and approved by the Institutional Review Board of the “Louis Turcanu” Children’s Emergency Hospital, Timisoara, Romania, under protocol number 20637/15.DEC.2023. To protect patient confidentiality and comply with ethical standards, personal data were not recorded or included in the study analysis. This ensured that the focus remained on the clinical and epidemiological aspects of *B. henselae* infections in the pediatric population while safeguarding individual privacy.

### 2.2. Inclusion Criteria

The absence of an age filter was deemed appropriate due to the hospital’s pediatric focus. The criteria for diagnosing systemic *B. henselae* infection were carefully defined to ensure accurate identification and analysis of cases. The diagnostic criteria included (1) contact with animals: any history of being in contact with animals or fleas, regardless of the presence or absence of bite marks, was considered a significant risk factor for *B. henselae* infection; (2) prolonged fever of unknown origin: patients presenting with fever that could not be attributed to any known cause over an extended period were evaluated for possible *B. henselae* infection; (3) hematological manifestation and/or hepatosplenic disease: any hematological abnormalities or evidence of liver or spleen disease, as determined through laboratory and imaging techniques, were critical indicators; (4) positive serological tests for *B. henselae*: confirmation of infection was based on positive serological tests for *B. henselae*, including the detection of specific IgM and IgG antibodies.

### 2.3. Demographical and Clinical Data

The following demographical data were collected: age, sex, parental education, family income, exposure to animals. Available clinical data were collected and included information on fever, lymphadenopathies, gastrointestinal symptoms, headaches, and used treatment options.

### 2.4. Laboratory Analysis and Imaging

We aimed to systematically review the laboratory and ultrasound imaging data from pediatric patient registries to identify key diagnostic variables related to systemic *B. henselae* infection. Our analysis focused on detecting morphological changes in the spleen using ultrasound imaging. We specifically looked for the presence of multiple hypoechoic lesions with clearly defined thick margins, a characteristic finding illustrated in Figure 1. Bone marrow aspiration was included as part of our diagnostic protocol to rule out differential diagnoses such as lymphoma.

As provided by our laboratory, analytes and their normal values included hemoglobin (mg/dL, 9.5–14.1 for children up to 12 years, 13.5–17.5 for adolescent boys and 12–16 for adolescent girls), white blood cell count (×10^3^ cells/ microliter, 6–17.5 for children up to 4 years, 5–14.5 for children up to 12 years and 4.5–11 for adolescents), platelets (×10^3^ cells/microliter, 150–450), erythrocyte sedimentation rate (ESR, mm/hour, 3–13 in children below 12, 3–20 in adolescents), C-reactive protein (CRP, mg/dL, 10), alanine aminotransferase (U/L, 7–56), and aspartate aminotransferase (U/L, 5–40).

Our serological evaluation was comprehensive, assessing HIV serology to exclude HIV infection, with all evaluated patients showing negative results. The tuberculin skin test and immunoglobulin levels were examined to ensure they fell within normal ranges, aiding in the diagnostic accuracy. Additionally, we screened for other conditions that could present with prolonged fever of unknown origin, such as sarcoidosis and cystic echinococcosis, which were not identified in the patients included in our study.

A significant aspect of our investigation was the identification of multiple infections through serological testing. The presence of antibodies against other infectious agents, including cytomegalovirus (IgM and IgG), Epstein–Barr virus (VCA IgM, IgG, and EBNA), and *Toxoplasma gondii* (IgG), was also documented. These serological results were crucial for understanding the complex infectious status of the patients and informed our diagnostic and treatment strategies. PCR testing and sequencing were later introduced in the clinic where the retrospective analysis took place.

## 3. Results

### 3.1. Background Characteristics

We examined the demographic and health characteristics of 19 pediatric patients diagnosed with systemic *B. henselae* infection over the specified period (Table 1). The median age of participants was 8.1 years, with an interquartile range (IQR) of 5.4 to 12.1 years, indicating a relatively young cohort. The age distribution showed that the majority of the cases were concentrated in the younger age groups, with 21.1% aged 0–4 years, 36.8% aged 5–9 years, 31.6% aged 10–14 years, and a smaller fraction, 10.5%, aged 15–18 years. Gender distribution was nearly balanced, with females slightly outnumbering males, comprising 52.6% and 47.4% of the cases, respectively.

Parental education levels varied, with half of the participants’ parents having an elementary level of education (52.6%), and the remainder split between having a high school education (26.3%) and holding a university degree (26.3%). Family income was reported as predominantly middle class for a significant majority of the cases (78.9%), with a smaller proportion classified as poor (15.8%) and rich (5.3%).

Exposure to felines was reported in 94.7% of the cases, with 42.1% having been exposed to cats and 52.6% to kittens. This high rate of exposure underscores the importance of cats, especially kittens, as vectors for the transmission of the bacterium to humans. Only one case (5.3%) had unknown feline exposure status. Other animal exposures were less common, with dogs being the next most frequently reported animal exposure (15.8%). A significant number of participants (63.2%) reported no exposure to animals other than cats or kittens, highlighting the specific risk posed by feline exposure in the transmission of *B. henselae*.

### 3.2. Clinical Findings

The analysis of clinical findings revealed that fever was the most common symptom, reported in 47.4% (9/19) of the cases, with a median duration of 5 days (IQR 2–12 days), indicating that nearly half of the affected children experienced fever, which persisted for several days, reflecting the systemic nature of the infection (Table 2). Chills and night sweats were less commonly reported, each observed in 5.3% (1/19) of the cases, with both symptoms lasting a median of 3 days. Notably, weight loss was not reported in any of the cases, indicating that this symptom may be less associated with systemic *B. henselae* infection in children. Headache and abdominal pain were each reported in 10.5% (2/19) of the cases, with headaches lasting a median of 8 days (IQR 4–14 days), and abdominal pain for a median of 3 days (IQR 1–4 days), suggesting that while these symptoms were relatively uncommon, they could persist and significantly impact the affected children.

Lymphadenopathy was the most prevalent finding, observed in 78.9% (15/19) of the cases, with a median duration of 9 days (IQR 5.5–21 days), underscoring the significant role of lymphadenopathy in systemic *B. henselae* infections, reflecting the organism’s predilection for lymphoid tissue. Among the specific types of lymphadenopathy, neck lymphadenopathy was the most common (31.6%), followed by axillary (21.1%), other (21.1%), and inguinal lymphadenopathy (10.5%). Splenomegaly and hepatomegaly were each observed in 5.3% (1/19) of the cases, highlighting that enlargement of the spleen and liver, while not widespread among the patients, can be significant findings in systemic infections.

### 3.3. Laboratory Findings

Our analysis revealed a mean hemoglobin level of 12.6 mg/dL, indicating that, on average, the children did not suffer from significant anemia. The mean white blood cell (WBC) count was elevated at 13.1 × 10^3^ cells per microliter, and the mean platelet count was 340.6 × 10^3^ per microliter, both within normal ranges but suggestive of inflammatory or infectious processes in some cases (Table 3). The erythrocyte sedimentation rate (ESR) varied widely among the patients, ranging from 3 to 140 mm/h, and the C-reactive protein (CRP) levels ranged from 0.2 to 29.4 mg/dL.

Elevations in liver enzymes were observed in 10.5% (2 of 19) of the children, pointing to hepatic involvement in a subset of patients. Leukocytosis and elevation in ESR were each present in 47.4% (9 of 19) of the cases, highlighting the common inflammatory response in systemic bartonellosis. Elevation in CRP was noted in 21.1% (4 of 19) of the children, further corroborating the presence of acute inflammation.

Serology revealed that 63.2% (12 of 19) of the children tested positive for *B. henselae* IgM antibodies, indicating recent or acute infection, while a significantly higher 94.7% (18 of 19) tested positive for IgG antibodies, suggesting past exposure or ongoing immune response to the infection.

Histopathological examination was conducted in 10.5% (2 of 19) of the cases, with one case (5.3%) showing necrotizing granulomatous inflammation, a finding consistent with Bartonella infection. No cases showed normal lymph nodes or positive Warthin–Starry stain, which is often used to visualize Bartonella bacteria, indicating the diagnostic challenge of confirming bartonellosis histopathologically. PCR testing for *B. henselae* was performed in one case (5.3%), with a 100% positivity rate.

### 3.4. Imaging Findings

A significant majority, 68.4% (13/19), underwent some form of radiological study, highlighting the reliance on imaging for diagnosis and assessment of systemic involvement (Table 4). Ultrasound (US), computed tomography (CT), and magnetic resonance imaging (MRI) were utilized in 94.7% (18/19) of the cases. Only a small fraction, 5.3% (1/19), had a chest radiograph alone.

Cervical lymphadenopathy was the most common finding, observed in 94.7% (18/19) of the children, which aligns with the known propensity of *B. henselae* to affect lymphatic tissue. Similarly, high frequencies of upper and lower extremity lymphadenopathy were each noted in 89.5% (17/19).

Abdominal imaging was performed in 26.3% (5/19) of the cohort, revealing abdominal lymphadenopathy in two cases. Notably, instances of splenomegaly and splenic or hepatic microabscesses were reported at a rate exceeding the number of abdominal imaging studies conducted.

Neuroimaging was undertaken in 15.8% (3/19) of the children, with all three cases showing findings suggestive of encephalitis, and an improbably high rate of optic neuritis was reported. Parotitis, identified through neck CT, was present in 21.1% (4/19) of cases, and bone involvement, excluding neuroimaging, was observed in 36.8% (7/19), indicating the infection’s capacity to affect diverse organ systems and tissues.

On imaging findings, we identified characteristic signs of systemic bartonellosis, as presented in Figure 2. The images revealed a profound impact of the infection across multiple organs. Specifically, Figure 2a demonstrates hepatic lesions that are distinguished by their central necrosis and surrounding diffuse edema, illustrating the severe effect of the infection on the liver. Figure 2b similarly depicts lesions within the spleen, following the same pattern of central necrosis enveloped by diffuse edema. These findings underscore the extensive reach of bartonellosis within the patient, affecting not only the liver and spleen but also extending to the skeletal system, including the right iliac bone, femoral neck, pubic bone, and L1–L2 vertebrae.

In the follow-up assessment of some patients using contrast-enhanced ultrasound (CEUS), significant findings were observed that provided deeper insights into the progression and response to treatment of Bartonella infections. One-month post-treatment, CEUS was performed, revealing a spleen of 9.6 cm. During the arterial phase of imaging, the spleen demonstrated uniform enhancement, characterized by a unique geographic pattern. Notably, this pattern included two distinct non-enhanced regions, one approximately 8 mm in size located at the lower pole and another measuring roughly 7–8 mm near the hilum. These CEUS findings, as illustrated in Figure 3, suggested the presence of splenic abscesses potentially in a state of healing, contrasting with earlier ultrasound results. The homogeneous enhancement across the spleen, punctuated by these well-defined round/oval non-enhancing areas, raised the possibility of a recuperative process underway within the organ, indicating a positive response to the administered treatment.

### 3.5. Treatment and Outcomes

All patients (100%) received antibiotic therapy, indicating a consistent approach to managing the infection across the cohort (Table 5). The majority, 84.2% (16/19), were initially prescribed antibiotics known to be active against Bartonella. Adjustments to the antibiotic regimen were made in 15.8% (3/19) of the cases. Azithromycin, either as a monotherapy or part of combination therapy, was the most frequently utilized antibiotic, used in 94.7% (18/19) of cases, underscoring its prominence in the treatment of bartonellosis. Specifically, azithromycin was used as a single drug in 73.7% (14/19) of cases. The combinations of azithromycin with rifampin, doxycycline, and gentamicin were less common, employed in 10.5%, 5.3%, and 5.3% of cases, respectively.

The median duration of treatment was 4 days, with an interquartile range of 4 to 10 days. Follow-up visits were recorded for 36.8% (7/19) of the patients, with 57.1% (4/7) of those followed up experiencing resolution or improvement of symptoms. There were no reported mortalities among the patients.

Co-infections were documented in a significant portion of the cohort, with Epstein–Barr Virus (VCA IgM, IgG, EBNA) present in 89.5% (17/19) of cases, indicating a high prevalence of this virus among the children studied. Cytomegalovirus (IgM and IgG) and Toxoplasma gondii (IgG) co-infections were identified in 15.8% (3/19) and 26.3% (5/19) of cases, respectively.

## 4. Discussion

These findings underscore the pediatric nature of systemic *Bartonella henselae* infections, with a significant prevalence in children under 14 years of age. The data also suggest that while the condition affects children across various socioeconomic backgrounds, exposure to young cats or kittens is a significant risk factor. This study provides valuable insights into the demographic and health characteristics of children with systemic *B. henselae* infection, offering a foundation for further research and targeted preventive measures.

The laboratory findings from our study underscore the varied hematological and serological manifestations of systemic *B. henselae* infection in children. The high rate of seropositivity for *B. henselae* IgG antibodies indicates widespread exposure among the patients, while the presence of leukocytosis and elevated ESR in nearly half of the cases reflects the systemic inflammatory response associated with this infection. The data highlight the need for comprehensive diagnostic approaches, including serology and, where possible, molecular diagnostics, to effectively identify and manage systemic bartonellosis in pediatric populations.

Bartonella infection or cat scratch disease (CSD) predominantly affects children and young adults [12]. On the other hand, new information suggests that adult prevalence may exceed earlier estimates. Granulomatous nonangiogenic inflammation combined with angiogenesis is a hallmark of CSD. This includes a neutrophilic inflammatory response to bacilli that are found in the skin, bone, and other organs; this can result in diseases such as bacillary angiomatosis or peliosis in the liver and spleen [12]. Atypical CSD presentations are less frequent in younger children, and diagnosing it becomes particularly challenging without evident lymphadenopathy. A U.S.-based study spanning from 2005 to 2014 highlighted that out of 224 reported cases, merely 36.2% were identified in children below 14 years of age [2].

In immunocompetent individuals, prolonged fever exceeding 3 weeks, accompanied by myalgia, arthralgia, and occasional skin eruptions, can heighten suspicions of a Bartonella infection [13,14]. Hepatic lesions have been reported in juvenile cases and, in rare instances, in adult cases, with or without concomitant splenic abnormalities detected by CT and ultrasonography scans [15,16]. Even in patients, especially children, who have classic symptoms of CSD but do not have overt systemic signs, hepatosplenic involvement may become apparent [16,17]. Such hepatosplenic engagements are identified in roughly a quarter of atypical CSD instances, with children below 14 years facing an elevated risk (relative risk (RR) 1.76, 95% confidence interval (CI) 1.04–2.99) [18]. Consequently, Bartonella serology becomes imperative when hepatosplenic nodules coexist with fever, as in our patients.

While isolated splenic CSD remains exceedingly uncommon, sporadic cases have been documented to date [19,20,21]. Given its resemblance to splenic lymphoma, a meticulous differential diagnosis becomes paramount [22]. Given the diverse clinical manifestations of CSD in immunocompetent individuals, it becomes crucial to discern and rule out other grave conditions that can present similarly to CSD.

Laboratory results for this condition are often nonspecific; infections might lead to leukocytosis, with ESR and CRP levels either within the normal range or elevated. Additionally, mild anemia and heightened liver enzyme levels can be observed in certain cases exhibiting systemic involvement [23]. A universally accepted gold standard for definitive diagnosis remains elusive. While Margileth proposed four diagnostic criteria, their practical application is challenging due to issues such as the complexities of performing biopsies and the limited sensitivity and specificity of serological tests, encompassing both EIAs and indirect immunofluorescence assays. The following criteria must be met: (1) contact with a cat or flea, with or without visible scratch marks or a localized inoculation lesion; (2) negative results from tests that rule out other possible causes of adenopathy or infection, such as PCR tests on blood or tissue samples and liver and spleen CT scans; (3) a serology test, such as an enzyme immunoassay (EIA) or indirect fluorescent antibody assay, showing a *B. henselae* titer greater than 1:64; (4) histopathological evidence from a biopsy of lymph nodes, skin, liver, or eye that demonstrates granulomatous inflammation consistent with CSD [18].

The diminished specificity of serological tests is attributed to their high seroprevalence in the general population, stemming from cross-reactivity with infections caused by *C. burnetii*, *Chlamydophila*, and non-*Bartonella henselae* strains [10,11]. Conversely, the diminished sensitivity is closely linked to the distribution of distinct *B. henselae* genotypes, further compounded by challenges in discerning previous infections. In the scenario presented, not all criteria set forth by Margileth could be met [18]. The diagnosis was initially suspected based on ultrasonographic findings of the spleen and liver, coupled with prolonged fever [24]. Multiparametric ultrasound, enriched with advanced Doppler techniques and contrast-enhanced capabilities, emerges as a viable, cost-efficient, and radiation-free modality for both diagnosing and monitoring hepatosplenic complications associated with systemic bartonellosis [25].

The management of disseminated CSD in immunocompetent adults remains largely empirical, with some cases resolving even without antibiotic intervention [26]. A pediatric study by Scolfaro et al. found that administering macrolides or a dual-antibiotic regimen for 2 to 3 weeks yielded swift clinical improvements [27]. Commonly recommended antibiotics include macrolides (such as erythromycin, azithromycin, and clarithromycin), fluoroquinolones (like ciprofloxacin), trimethoprim-sulfamethoxazole, and doxycycline [27]. Rifampin and gentamicin found utility in treating hepatosplenic CSD in children, as indicated by the research of Arisoy et al. [28].

In our cohort, one of the cases characterized by systemic manifestations involving the spleen, liver, and bones received treatment with a combination of antibiotics for 6 weeks. Conversely, the second case, presenting milder symptoms, was managed solely with oral monotherapy for 4 weeks. Interestingly, another third case, despite showing systemic signs like splenic lesions, exhibited a rapid recovery when treated with azithromycin alone. The decision to opt for azithromycin was swayed by the patient’s tender age, wherein the use of certain other antibiotics is cautioned against. A 2004 guideline suggests combining doxycycline (100 mg PO or IV twice daily) with rifampin (300 mg PO twice daily) for addressing complex CSD cases. However, the optimal treatment duration for immunocompetent individuals with intricate CSD remains undefined [26].

### Limitations

The retrospective design inherently limits our ability to capture all relevant data comprehensively and accurately, potentially leading to information bias. The small sample size of 19 pediatric patients may not fully represent the broader population affected by this condition, thereby limiting the generalizability of our findings. Furthermore, the reliance on hospital records from a single geographical region restricts our ability to extrapolate results to other settings or populations with different epidemiological profiles. Additionally, the diagnostic criteria, including serological testing, may be influenced by cross-reactivity with other pathogens, introducing diagnostic uncertainty. The identification of co-infections was based on available serological data, which might not reflect the full spectrum of concurrent infections due to the selective testing approach.

## 5. Conclusions

Diagnosing systemic bartonellosis in children, caused by *B. henselae*, remains complex due to its varied clinical presentations, ranging from asymptomatic cases in immunocompetent individuals to localized or systemic manifestations such as regional adenopathy and prolonged fevers of unknown origin. These prolonged fevers present significant diagnostic challenges, underscoring the necessity for healthcare professionals to recognize diverse clinical signs, including systemic features like splenic anomalies, for timely and accurate diagnosis. However, the diagnostic challenge is heightened by the occurrence of co-infections with other pathogens, which may mask or mimic the symptoms of *B. henselae*, complicating clinical assessments and management decisions. Early detection is crucial for initiating prompt and effective treatment. Pediatric treatment strategies must navigate age-related antibiotic considerations, often leading to the careful selection of therapies like azithromycin monotherapy, which can offer swift and favorable outcomes. The management of systemic bartonellosis in children calls for a comprehensive approach that combines clinical evaluation, advanced imaging techniques, and carefully chosen antibiotic treatments tailored to each child’s specific needs, highlighting the importance of a holistic and nuanced approach in pediatric infectious disease management.

## Figures and Tables

**Figure 1 microorganisms-12-00666-f001:**
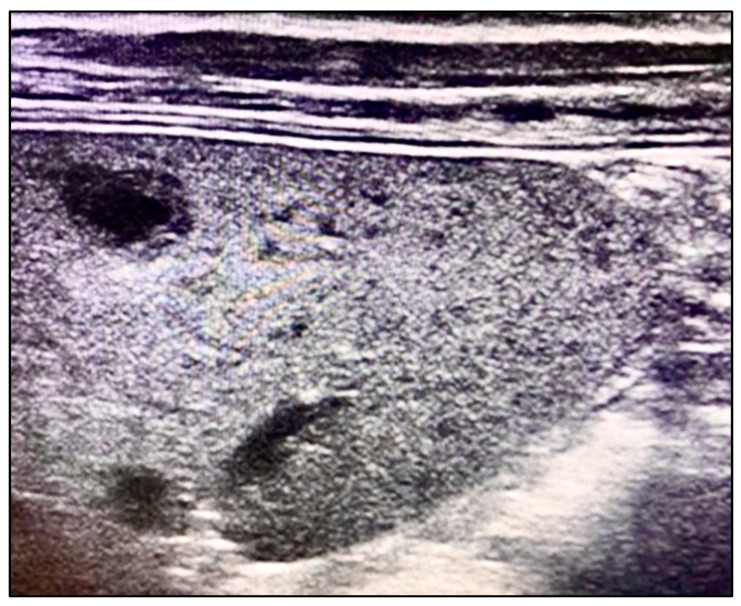
Multiple hypoechoic lesions with well-defined thick margins on ultrasound.

**Figure 2 microorganisms-12-00666-f002:**
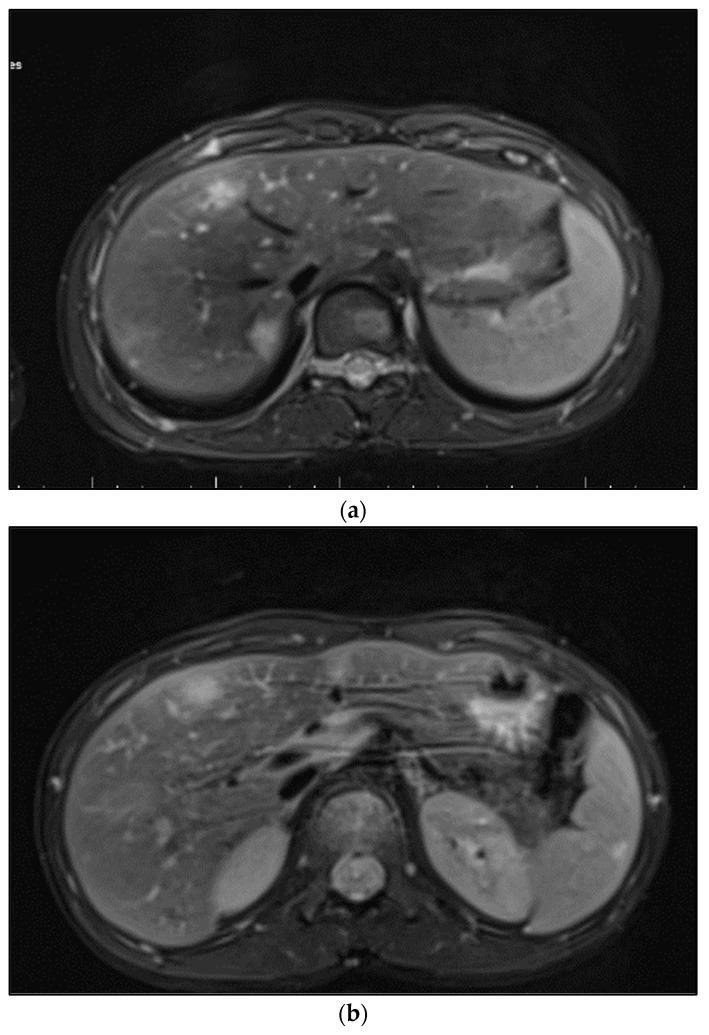
(**a**) Hepatic lesions with central necrosis and diffuse edema; (**b**) spleen lesions with central necrosis and diffuse edema.

**Figure 3 microorganisms-12-00666-f003:**
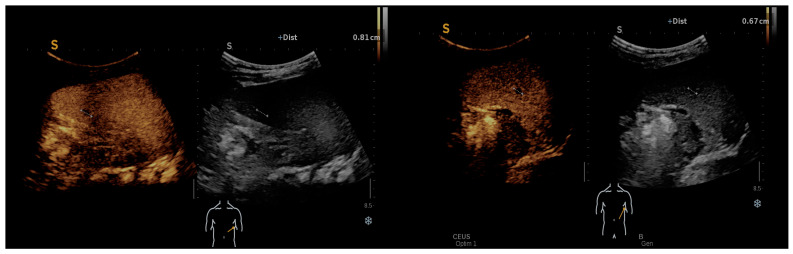
CEUS: homogeneous enhancement revealing two well-defined splenic round/oval non-enhancing areas.

**Table 1 microorganisms-12-00666-t001:** Demographic and health characteristics of the study participants.

Variables	N = 19	%
Age, years (median, IQR)	8.1 (5.4–12.1)	0.502
**Age group**		
0–4 years	4	21.1
5–9 years	7	36.8
10–14 years	6	31.6
15–18 years	2	10.5
**Gender**		
Female	10	52.6
Male	9	47.4
**Parents’ education**		
Elementary	10	52.6
High school	5	26.3
University degree	5	26.3
**Family income**		
Poor	3	15.8
Middle class	15	78.9
Rich	1	5.30
**Feline exposure**		
Cats	8	42.1
Kittens	10	52.6
Unknown	1	5.30
**Other animal exposures**		
Dog	3	15.8
None	12	63.2
Others	1	5.30
Unknown	3	15.8

IQR—Interquartile Range.

**Table 2 microorganisms-12-00666-t002:** Clinical findings.

Variables	Frequency	%
**Fever**	9 of 19	47.4
Fever duration, days (median, IQR)	5 (2–12)	-
**Chills**	1 of 19	5.30
Chills duration, days (median, IQR)	3 (2.5–6)	-
**Night sweats**	1 of 19	5.30
Night sweats duration, days (median, IQR)	3 (3–6)	-
**Weight loss**	0 of 19	0.0
Weight loss duration, days (median, IQR)	1 (1–3)	-
**Headaches**	2 of 19	10.5
Headache duration, days (median, IQR)	8 (4–14)	-
**Abdominal pain**	2 of 19	10.5
Abdominal pain duration, days (median, IQR)	3 (1–4)	-
**Lymphadenopathies**	15 of 19	78.9
Lymphadenopathy duration, days (median, IQR)	9 (5.5–21)	-
Neck lymphadenopathy	6 of 19	31.6
Axillary lymphadenopathy	4 of 19	21.1
Inguinal lymphadenopathy	2 of 19	10.5
Other lymphadenopathy	4 of 19	21.1
Splenomegaly	1 out of 19	5.30
Hepatomegaly	1 out of 19	5.30

IQR—Interquartile Range.

**Table 3 microorganisms-12-00666-t003:** Laboratory findings in children with systemic bartonellosis.

Variables	Frequency/Mean	Standard Deviation
**Blood Counts**		
Mean Hemoglobin	12.6 mg/dL	0.6
Mean White Blood Cell Count	13.1 × 10^3^ cells/mL	4.3
Leukocytosis	9 of 19 children (47.4%)	
Mean Platelet Count	340.6 × 10^3^ per microliter	12.3
ESR	19.1 mm/h	11.1
Elevation in ESR	9 of 19 children (47.4%)	
**Biochemistry**		
CRP	7.9 mg/dL	7.1
Elevation in CRP	4 of 19 children (21.1%)	
Elevation in Liver Enzymes	2 of 19 children (10.5%)	
**Other**		
Serology for *B. henselae* (IgM Positive)	12 of 19 children (63.2%)	
Serology for *B. henselae* (IgG Positive)	18 of 19 children (94.7%)	
Cases with Histopathology	2 of 19 cases (10.5%)	
Necrotizing Granulomatous Inflammation	1 of 19 cases (5.3%)	
Normal Lymph Node	0 of 19 cases (0%)	
Positive Warthin–Starry Stain	0 of 19 cases (0%)	
PCR Testing for *B. henselae*	1 of 19 cases (5.3%)	
PCR Positive for *B. henselae*	1 of 1 PCR tested cases (100%)	

**Table 4 microorganisms-12-00666-t004:** Imaging findings in children with systemic bartonellosis.

Variables	Frequency	Percentage (%)
Radiological Study Available	13 of 19	68.4
Chest Radiograph Only	1 of 19	5.3
Advanced Imaging (US, CT, MRI)	18 of 19	94.7
Concurrent Disease Process	1 of 19	5.3
Cervical Lymphadenopathy	18 of 19	94.7
Upper Extremity Lymphadenopathy	17 of 19	89.5
Lower Extremity Lymphadenopathy	17 of 19	89.5
Abdominal Imaging Performed	5 of 19	26.3
- Abdominal Lymphadenopathy	2 of 5	
- Splenomegaly	7 of 5	
- Splenic and/or Hepatic Microabscesses	7 of 5	
Neuroimaging Performed	3 of 19	15.8
- Findings Suggestive of Encephalitis	3 of 3	
- Optic Neuritis	5 of 3	
Parotitis (Neck CT)	4 of 19	21.1
Bone Involvement (excluding neuroimaging)	7 of 19	36.8

**Table 5 microorganisms-12-00666-t005:** Treatment and outcomes.

Variables	Frequency	Percentage (%)
**Antibiotic Use**	19 of 19	100
Initial Antibiotics Active Against Bartonella	16 of 19	84.2
Antibiotics Adjusted	3 of 19	15.8
Azithromycin Used as Single Drug	14 of 19	73.7
Azithromycin in Combination	18 of 19	94.7
- with Rifampin	2 of 19	10.5
- with Doxycycline	1 of 19	5.3
- with Gentamicin	1 of 19	5.3
Median Treatment Duration (Days)	4 (IQR, 4–10)	
Follow-up Visits	7 of 19	36.8
Resolution or Improvement at Follow-up	4 of 7	57.1
Mortality	0 of 19	0
**Documented Co-infections**		
- Cytomegalovirus (IgM and IgG)	3 of 19	15.8
- Epstein–Barr Virus (VCA IgM, IgG, EBNA)	17 of 19	89.5
- *Toxoplasma gondii* (IgG)	5 of 19	26.3

## Data Availability

The raw data supporting the conclusions of this article will be made available by the authors on request.

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
