# Peer review of "A Retrospective Analysis of Systemic Bartonella henselae Infection in Children"

_microorganisms, 2024, doi:10.3390/microorganisms12040666_

Round 1

Reviewer 1 Report

Comments and Suggestions for Authors

Dear authors, congratulations on your work. I have some considerations about it.

Introduction Section: It should include information about the etiological agent. Currently, there is limited information about the pathogen and the disease, as well discussed in the paragraph on Line 302-311.

Line 62: Instead of using abbreviations, please write out “CT” and “MRI”.

Line 71: The statement “An unchecked atypical Bartonella infection can develop over a few days [9]” is unclear. Please provide clarification.

Justification: Consider adding a justification for the work at the end of the Introduction section.

Inclusion Criteria: Specify whether the inclusion criteria include only the listed characteristics or additional ones. Also, clarify whether children need to meet one or more of the criteria.

Figure 1: Evaluate whether Figure 1 is necessary for the paper.

Lines 119-132: Separate the description of tests used from the results. If positive and negative results were essential for patient inclusion, describe this in the Inclusion criteria.

Tables: Add subdivisions (lines or bold) to define different subtopics or information within the tables.

Material and Methods: Update this section to include a brief description of demographics and clinical information obtained.

Results and Tables: Use “per” instead of “/” to avoid redundancy.

Table 3: Present data with upper and lower limits, numerical values, and complete information. Avoid vague terms like “elevation,” “range,” “normal,” “positive,” and “negative.”

Discussion: Consider whether some of the review data could be moved to the Introduction section.

Author Response

Thank you for your support. Please find the answers attached.

Reviewer 2 Report

Comments and Suggestions for Authors

Introduction: I recommend that the authors include more specific information about B. henselae itself. This may include detailed information on the characteristics, transmission and epidemiology of B. henselae. Including these details would strengthen the introduction and improve the overall clarity of the article.

Editing suggestions:

Line 28: write in Italic Bartonella henselae.

Lines 32, 34, 38, 44, 48 and 53: write in Italic B. henselae.

Lines 47 and 129: write in Italic Toxoplasma gondii.

Lines 66 and 71: write first letter capital and in Italic word bartonella.

Lines 89, 94, 100, 103,104, 109, 123, 137, 156, 168, 175, 199, 207, 218, 287, 291, 295, 296, 345, 386 and 394: abbreviate Bartonella henselae.

Line 98: abbreviate and write in Italic Bartonella henselae.

Line 149: delete "... a known risk factor for Bartonella henselae infection..." because it is not your results.

Lines 206, 233, 250, 268, 302, 314, 320: write in Italic Bartonella.

Line 282: abbreviate and write in Italic Toxoplasma gondii.

Table 1-5: has no reference in the text.

Line 345: abbreviate Coxiella burnetii.

Author Response

(The authors gave the same response as above.)

Round 2

Reviewer 1 Report

Comments and Suggestions for Authors

Thanks for the review.